# A novel DNA repair-independent role for Gen nuclease in promoting unscheduled polyploidy cell proliferation

Manon Budzyk[1], Anthony Simon [1], Anne-Sophie Mace[2], Renata Basto [1]*

1 Biology of Centrosomes and Genetic Instability Lab, Institut Curie, PSL Research University, CNRS UMR 144, Paris, France, 2 Cell and Tissue Imaging Facility (PICT-IBiSA), Institut Curie, PSL Research University, CNRS, Paris, France

* renata.basto@curie.fr

## Abstract

Unscheduled whole genome duplication (WGD), also described as unscheduled or non-physiological polyploidy, can lead to genetic instability and is commonly observed in human cancers. WGD generates DNA damage due to scaling defects between replication factors and DNA content. As a result DNA damage repair mechanisms are thought to be critical for ensuring cell viability and proliferation under these conditions. In this study, we explored the role of homologous recombination and Holliday junction resolution in non-physiological polyploidy *in vivo*. Using *Drosophila* genetics and high-resolution imaging, we identified a key and surprising role for Gen/Gen1 nuclease. Our findings revealed that loss-of-function and overexpression of Gen have opposing effects, delaying or accelerating the proliferation of polyploid cells, respectively. These changes ultimately impact cell proliferation, nuclear asynchrony and mitotic DNA damage levels. Surprisingly, our findings show that this effect is unrelated with the expected Gen's function in DNA damage repair. Instead, Gen seems to influence polyploid DNA replication rates. This work identifies a novel function for Gen nuclease and provides new insights into the cellular and molecular requirements of non-physiological polyploidy.

## Author summary

Unscheduled polyploid cells experience replication stress that leads to DNA damage. The proliferation of such damaged cells typically depends on robust DNA damage repair mechanisms such as homologous recombination (HR), an error-free repair pathway. In this study, we investigated the role of proteins involved in HR in suppressing unscheduled polyploid cell proliferation. Unexpectedly, we found that one protein—the nuclease Gen—plays an essential role in promoting the proliferation of unscheduled polyploid cells, while other HR proteins appear to be dispensable. Notably, Gen is not required for the

**Data availability statement:** All data are in the manuscript and/or Supporting information files.

**Funding:** This work was supported by the European Research Council Consolidator Grant (ChromoNumber-LS3, ERC-2016-COG) to R.B. and the French government for the ScalingRules project from ANR chair excellence through the Agence Nationale de la Recherche from France 2030 (ANR-23-CHBS-0012) to R.B, the InCA (www.e-cancer.fr) (2021-1- PREV-Bio grant to R.B.), the Institut Curie and the Centre National de la Recherche Scientifique. M.B.'s salary was funded by the ministry of Science and the ARC foundation. The funders had no role in study design, data collection and analysis, decision to publish, or preparation of the manuscript.

**Competing interests:** The authors have declared that no competing interests exist.

proliferation of diploid cells even if replication stress is induced through drug treatments. Although Gen has been primarily characterized as a nuclease functioning in the HR pathway, our findings reveal that its role in supporting polyploid cell proliferation is independent of both its nuclease activity and its function in the DNA damage response. These results uncover a novel, non-canonical role for Gen protein in promoting cell proliferation under conditions of genetic instability

## Introduction

Polyploidy, resulting from WGD is defined as the presence of more than two copies of all chromosomes. Several cellular strategies can be used to obtain polyploid cells, such as endoreplication, cell fusion and endomitosis, which includes mitotic slippage and cytokinesis failure [1,2]. In mammals, certain organs contain polyploid cells, such as cardiomyocytes and hepatocytes in the heart and liver, respectively, contributing to essential physiological roles [1]. A protective role for polyploidy in the ageing *Drosophila* brain has recently been shown [3]. Further, polyploid neurons have been described in the mouse neocortex during embryonic development [4]. In sharp contrast to scheduled polyploidy, unscheduled or non-programmed polyploidy resulting from whole genome duplication (WGD) is not associated with a protective role. Indeed, this is the second most frequent alteration in human cancers leading to high levels of genetic instability that fuel tumor evolution [5–9]. In these conditions, replicative stress and high levels of DNA damage can be generated in a single S-Phase [10]. Ensuring appropriate DNA damage repair is essential to allow a certain degree of viability even if leading to genetic instability.

In the presence of DNA double strand breaks (DSBs), cells activate a conserved DNA damage response (DDR), which involves different players that slow down or arrest the cell cycle to allow DNA damage repair [11,12]. Homologous recombination (HR) is one of the most efficient pathways involved in DSB repair and is activated later in the cell cycle as DNA is replicated [13]. It is an error-free DNA repair mechanism that uses the homologous chromosome as a template to repair DNA lesions. During HR, four-stranded DNA intermediates called Holliday Junctions (HJs) are formed and these need to be removed to ensure proper chromosome segregation during mitosis [14,15]. Three distinct pathways have been described in HJ processing. One involves Bloom helicase (BLM), topoisomerase IIIα and the RecQ-mediated genome instability proteins 1 and 2, which is known as the BTR complex. The second, requires several nucleases: SLX1–SLX4–MUS81–EME1- the SLX–MUS complex [16]. Finally, the third mechanism involves a single nuclease called Gen or GEN1 in humans [17,18]. The processing pathway choice appears to be cell cycle dependent, with BTR preferentially used during S phase, the SLX-MUS complex in G2 and GEN1 in late G2 and mitosis [19]. Indeed, human GEN1 has a nuclear export sequence and localizes preferentially in the cytoplasm, which suggests that HJ intermediates are processed by GEN1 after nuclear envelope breakdown [20,21]. In yeast, the activity of Yen1, the GEN homolog, is regulated via

Cdc14 dephosphorylation during mitosis, which allows nuclear localization and DNA binding [22,23]. More recently, it has been found that GEN is required to cleave under-replicated regions at specific loci called common fragile sites to facilitate Mitotic DNA Synthesis (MIDAS) [24]. *In vivo*, both GEN1 and Yen1 act redundantly to MUS81 nuclease. MUS81 loss results in hypersensitivity to a large range of DNA damaging agents, while mutations in GEN1/Yen1 do not cause DNA repair defects on their own but increase the severity of MUS81 loss [25–27]. Interestingly, this hierarchical relationship appears to be reversed in *Drosophila*, as Gen (the fly ortholog of GEN1) single mutants are more hypersensitive to DNA damaging agents than Mus81 single mutants [28]. In addition, *in vitro* studies showed that *Drosophila* Gen can cleave replication fork substrates, suggesting that it can also process DNA replication intermediates in flies [29].

Unscheduled polyploidy generates high levels of DNA damage in an DNA replication dependent manner. It is therefore reasonable to consider that HR must be at play in DDR in these cells. Accordingly, HR components were found to be specifically lethal in polyploid yeast [30] and Rad51 levels were increased in human tetraploid cells [10]. Here we investigated the role of HR and HJ proteins in unscheduled polyploid cell proliferation. We focus our study on Gen nuclease, as its depletion had a severe effect on cell cycle progression in polyploid NBs. Surprisingly, Gen's function appears to be dependent on DNA replication rates and it is uncoupled from its role in HR/HJ processing pathways and even from DNA damage repair in general.

## Results

### Rad51 and Gen impair polyploid cell proliferation

To identify DNA damage repair specific vulnerabilities of unscheduled polyploid proliferation, we focused on the homologous recombination (HR) pathway. We reasoned that HR should be involved in DNA damage repair since an essential role for this pathway in polyploid yeast has been previously shown [30]. In addition, in newly born unscheduled human tetraploid cells, DNA damage is generated during S-Phase [10], which relies on HR for repair before mitosis.

We used an *in vivo* system based on the developing *Drosophila* brain. Wild type brains contain neural stem cells, also known as neuroblast (NBs), which are diploid, i.e., contain two copies of all chromosomes. Mutations that affect mitosis can generate polyploid NBs that continue to proliferate accumulating a large number of all chromosomes (Smith et al., 1985; Karess et al., 1991; Reed and Orr-Weaver, 1997; Nano et al., 2019; Goupil et al., 2020) [31–35]. We used the hypomorphic mutant *spaghetti squash- sqh¹-* referred here as *sqh^mut* (Karess et al., 1991). The *sqh* gene encodes the non-muscle myosin II regulatory light chain, which is absolutely required for cytokinesis. In *sqh^mut* NBs, cytokinesis failure initially generates binucleated NBs, however as they continue to cycle and accumulate increasing chromosome numbers, large multilobated nuclei are frequently noticed (Fig 1B, 1C), [33,35]. In this system, genetic instability results from diverse events including replicative stress, cell cycle asynchrony or defects during mitosis [10,33,34]. Importantly, in these polyploid NBs, cell area correlates with nuclear area and so with DNA content (Fig 1D), allowing us to use cell area as a proxy for polyploidy levels in conditions of RNA interference (RNAi) (Fig 1E). In diploid wild type (WT) brains, NBs displayed smaller sizes, while in s*qh^mut* brains- here referred to as polyploid brains, a continuum of cell areas can be identified (Fig 1F, 1G). This is related with the hypomorphic nature of the s*qh* allele (different NBs will start to fail cytokinesis at different stages of development). Importantly, these differences inform on the extent of proliferation. Here, proliferation is used to describe an increase in cell area through an increase in DNA content, rather than the classical process describing the increase in cell number through cell division. A larger polyploid NB containing more nuclei than a smaller polyploid NB indicates that the larger cell has experienced more cell cycles than a smaller polyploid NB.

Using a NB specific GAL4 driver (Worniu-GAL4), we depleted by RNA interference (RNAi) Rad51, BLM, Mus81, SLX1 and Gen in polyploid brains. Rad51 is a major player in the early steps of HR [13], while BLM has both early and late functions [36,37]. BLM, Mus81, SLX1 and Gen function in Holiday Junction (HJ) processing [16,38]. HJs are byproducts of HR. Depletion of Rad51 decreased polyploid NB area Fig 1F, 1G), consistent with a crucial role of HR in polyploid cell proliferation. Characterization of polyploid, Rad51^RNAi brains, revealed an increase in their mitotic index (calculated as the

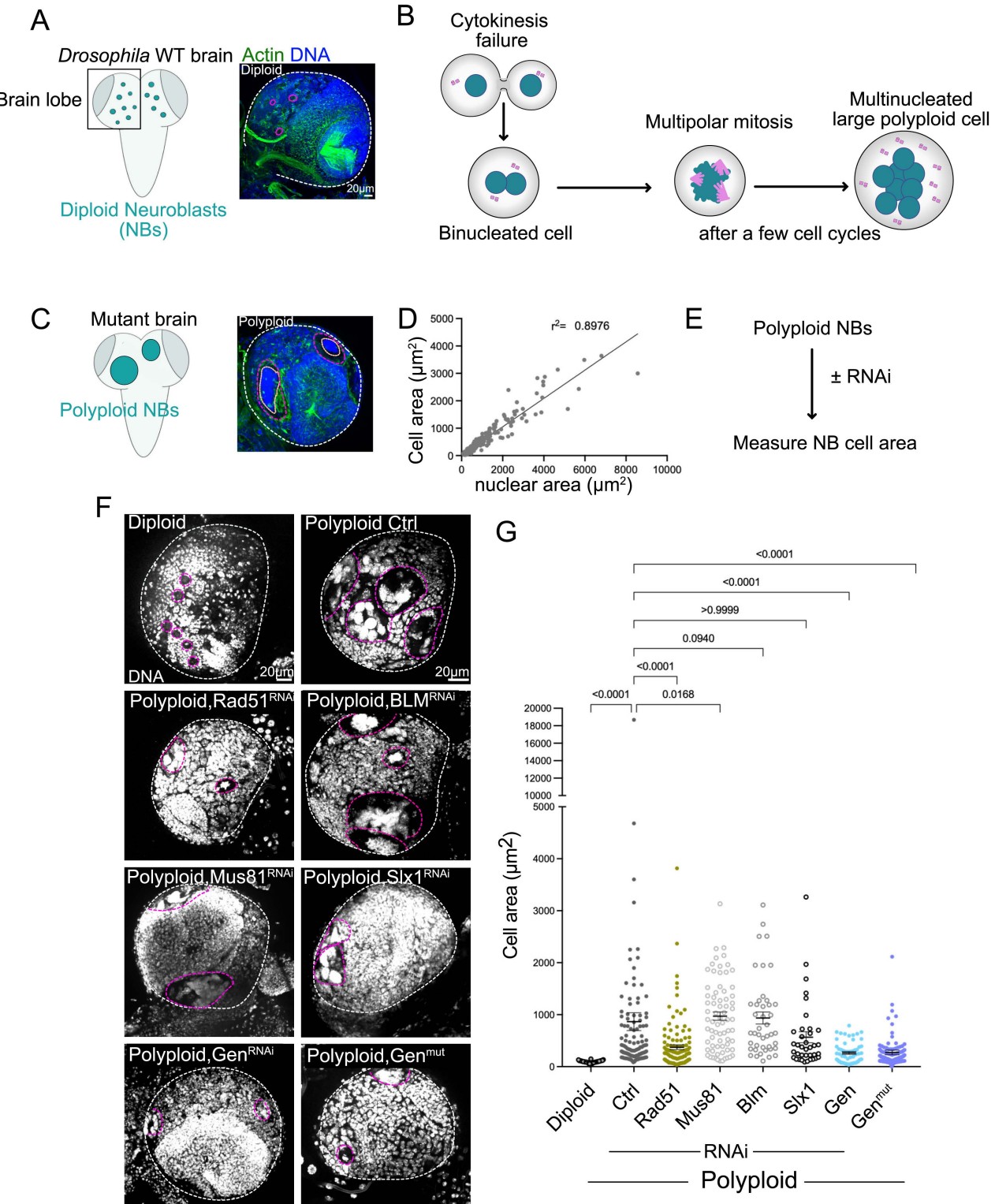

**Fig 1. Loss of Gen nuclease inhibits unscheduled polyploid NB proliferation.** (A) Left- drawing illustrating a WT Ctrl diploid brain depicting the neural stem cell population-called Neuroblasts (NBs); Right- Picture of a WT Ctrl diploid brain lobe showing Actin and DNA labeling. (B) Drawing illustrating the process of cytokinesis failure leading to multinucleated polyploid NBs. DNA is shown in green and centrosomes in pink. After a few rounds

of cytokinesis failure, multinucleated cells can be identified. (C) Left- drawing illustrating a polyploid *sqh*^*mut* brain; Right- Picture of a polyploid *sqh*^*mut* brain lobe showing Actin and DNA labeling. In A and C, Actin in green and DNA in blue. White dashed lines surround the brain lobes, while pink dashed lines surround individual NBs and in (C) orange dashed lines surround the nuclei. (D) Graph showing the correlation between cell and nuclear area of polyploid NBs, $r^2$ = Pearson correlation coefficient (p<<0,0001). (E) Diagram of the strategy used to characterize polyploid NB cell area. (F) Pictures of brain lobes of the indicated genotypes. DNA in grey. Note that in all these experiments, the NB marker DPN was used to identify NBs. The white dashed line surrounds the brain lobe and the pink dashed lines surround polyploid NBs (G) Dot plot showing cell area of the indicated genotypes. Bars show the mean ± SEM. Statistical significance is shown and determined by Kuskal-Wallis non-parametric test. Experiments were repeated at least two times with a minimum of 15 brain lobes analyzed per condition.

ratio between the number of NBs in mitosis and the total number of neuroblasts per brain lobe). In tissues, an increased mitotic index can be explained by cells spending more time in mitosis (or even by being arrested in mitosis), or alternatively, by presenting increased proliferation capacity. We favor the first possibility, as Polyploid, Rad51^RNAi NBs are smaller than polyploid NBs, suggesting a break in proliferation. Most likely, the increased DNA damage levels found in Polyploid, Rad51^RNAi NBs (S1A–S1C Fig) impair mitotic progression. In contrast to Rad51, depletion of BLM or Mus81 did not impact polyploid cell area, while Slx1 depletion cause polyploid cell area reduction, but this was not statistically significant (Fig 1F, 1G). Surprisingly, depletion of Gen, resulted in very small polyploid cells. This was further confirmed using a mutation in Gen, here referred to as Gen^mut (Fig 1F, 1G) (see methods).

### Gen loss-of-function mutant diploid brains show slower cell cycles

The role of Gen nuclease in sustaining polyploid cell proliferation was unexpected. We decided therefore to focus our work on this protein. Since little is known about Gen's function *in vivo* and in the *Drosophila* brain, we first characterized diploid Gen^mut brains. Analysis of third instar larval brains (Figs S2A and 2A) and in agreement with previous studies [39], revealed no obvious developmental defects. The number of NBs was similar between wild type control diploid brains (WT Ctrl), while their mitotic index was increased (S2B, S2C Fig). Time-lapse imaging approaches revealed that mitotic duration was extended in Gen^mut, during prometaphase and metaphase (S2D Fig). Further an increase in the duration of mitosis and of the overall cell cycle was also noticed (2B, 2C Fig). Characterization of mitotic defects in anaphase revealed an increase in the frequency of anaphases displaying lagging chromosomes, acentric chromosomes and chromosome bridges (S2E-2EF Fig). We concluded that Gen^mut NBs display mitotic defects and increased lengthening of mitosis and cell cycle timing.

### Over-expression of Gen nuclease in diploid NBs results in faster cell cycles

With the aim of further characterizing Gen function *in vivo*, we generated tagged versions of *Drosophila* Gen protein. We made transgenic flies expressing full length Gen coding region tagged with a fluorophore at the C-terminus (away from the nuclease domain which is localized on the N-terminus). We generated two different constructs with different promoters. The first one, used the ubiquitin (Ubq) promoter that drives a ubiquitous mild over-expression of Gen fused to NeonGreen (NeG) [40], as reported in other studies [41,42]- referred as mild GenOE. The second, - an UAS promoter fused to mCherry (mCh). This promoter generates high over-expression conditions in a tissue-specific manner [43] - referred to as strong GenOE (Figs 2D and S3A). In diploid NBs, both transgenes showed a strong cytoplasmic localization in interphase cells (Figs 2E and S3B), as described in human cells [20]. However, a lower signal in the nucleus was also detected in interphase NBs, as recently described in human cells [24]. Measuring Gen-mCh fluorescence intensity (FI) in diploid NBs confirmed the presence and extent of signals in the cytoplasm and the nucleus (Fig 2E, 2F). Time-lapse microscopy revealed that during mitosis, the cytoplasmic fluorescence signal spread throughout the cell, covering the chromosome and mitotic spindle region. Gen was subsequently excluded as the nucleus reformed in telophase (Fig 2G). We tested the functionality of our constructs by crossing the mild GenOE transgene with Gen^mut and analyzed the mitotic

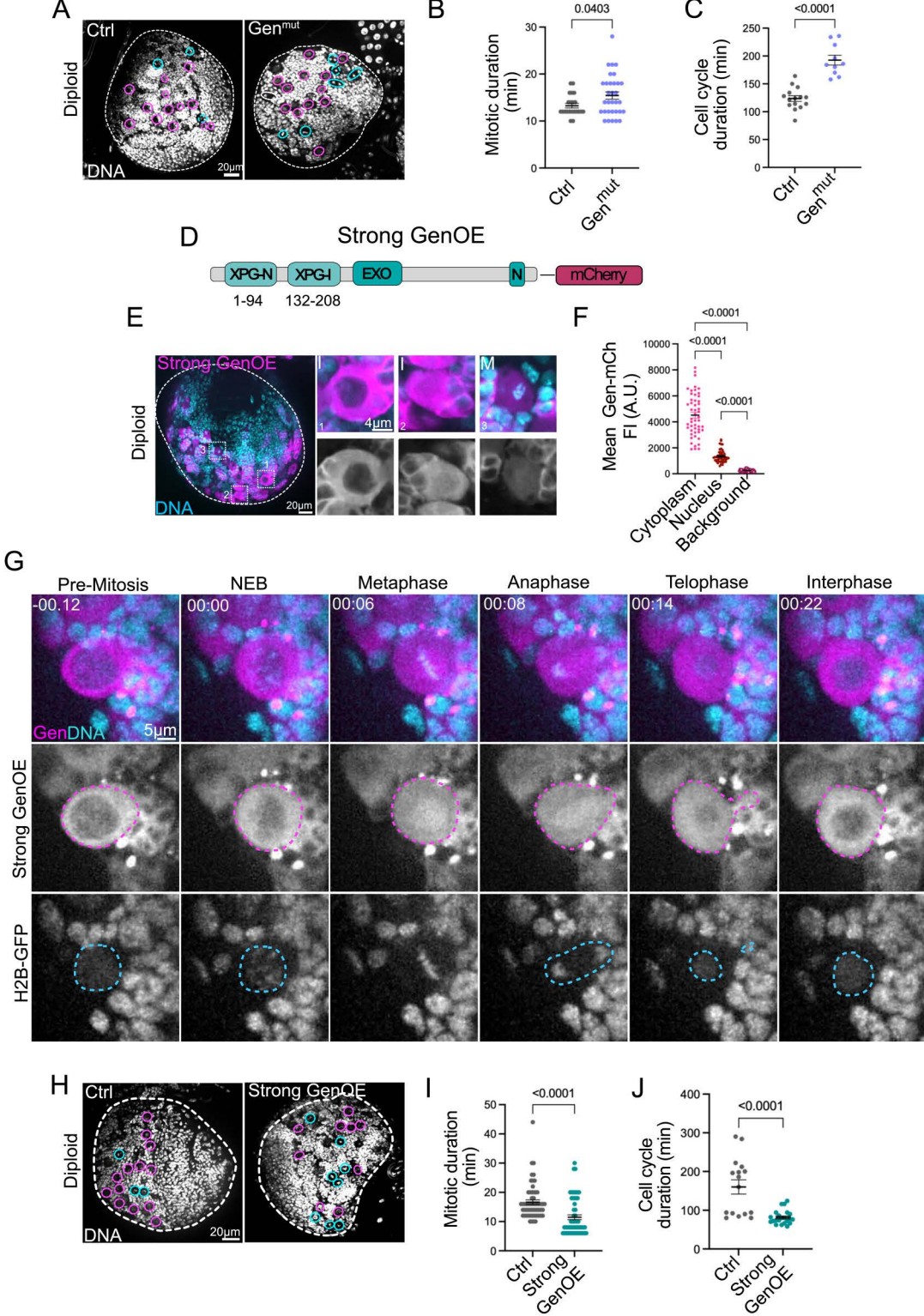

**Fig 2. Characterization of Gen^mut and GenOE diploid NBs reveal cell cycle timing alterations.** (A) Pictures of Ctrl (left) and Gen^mut (right) brain lobes showing DNA labelling. (B, C) Dot plot graphs showing mitotic and cell cycle duration of the indicated genotypes. (D) Diagram of the Strong GenoE transgene, where the full coding region was fused at the C-terminus with a mCherry (mCh) tag. (E) Left- brain lobe expressing the strong GenOE

transgene (pink). DNA is in cyan. Right- insets of interphase (I) and mitotic (M) NBs. Below GenOE is shown in grey. The numbered squares correspond to the inset numbers. Note that interphase NBs can show very low nuclear Gen levels (1) or higher Gen levels, while mitotic NBs show Gen spreading throughout the cytoplasm overlapping with chromosomes. (F) Dot plot graph showing Gen-mCh mean intensity values in the cytoplasm, nucleus and background- taken in positions from outside the brain region. (G) Stills of time lapse movies of NBs from third instar brains expressing strong GenOE-mCherry (in pink in the merged pictures at the top and in grey in the middle panel) and H2B-GFP (in cyan in the merged pictures at the top and in grey at the bottom). The pink dashed lines surround the NB cell and the blue dashed line the nucleus. Time is shown in minutes and time zero was defined at nuclear envelope breakdown. (H) Pictures of Ctrl (left) and strong GenOE (right) brain lobes showing DNA labelling (grey). For A, B and H the white dashed lines surround the brain lobes. In A and H, pink circles mark NBs in interphase, while blue circles mark mitotic NBs. (I, J) Dot plot graphs showing the mitotic duration (I) and cell cycle duration (J) in Ctrl and Strong GenOE NBs. B-C, F and I-J, Bars indicate the mean and SEM. Statistical significance is shown and determined by Two-tailed Mann-Whitney tests. Experiments were repeated at least two times with a minimum of 10 NBs from 10 different brain lobes analyzed per condition.

index. Interestingly, the combination of Gen$^{mut}$ and the Mild GenOE transgene, revealed a lower mitotic index than in Gen$^{mut}$ brains and even lower than in WT brains (although not statistically significant compared to Ctrl brains) (S3C Fig). These results show that the higher mitotic index of Gen$^{mut}$ is rescued by a single copy of the Gen transgene, attesting for its functionality.

Further characterization of diploid strong GenOE brains revealed similar NB numbers but an increased mitotic index, even higher than in Gen$^{mut}$ brains (Figs 2H and S3D, S3E). Using live imaging approaches, we determined the length of mitosis and of the cell cycle in strong and mild GenOE NBs. Strikingly, and in sharp contrast with Gen$^{mut}$ NBs, strong GenOE NBs displayed both faster mitoses and faster cell cycles (Fig 2I, 2J). Interestingly, the mild GenOE transgene did not cause this acceleration of cell division or cell cycle progression (S3F, S3G Fig). To confirm that the decrease in cell cycle duration was specific to strong GenOE, we also generated a *Drosophila* line overexpressing Mre11 (Mre11OE) - another nuclease involved in DNA damage signaling and repair [44]- using the UAS promoter. We measured mitotic duration in Mre11OE NBs but found that Mre11OE NBs had a similar mitotic duration compared to Ctrls (S3H, S3I Fig). This confirms that faster cell cycles are specific to Gen nuclease over-expression, and that overexpressing other members of a DNA repair pathway is not sufficient to accelerate mitosis or cell cycle progression.

Analysis of mitosis likewise revealed a variety of mitotic defects in GenOE anaphases, including lagging chromosomes, chromosomal bridges and acentric chromosomes (S3J Fig). Together, our findings suggest that Gen levels have to be tightly regulated during development to avoid defects in cell cycle progression, mitotic timing and mitotic fidelity.

**Gen is dispensable for DNA damage repair in diploid NBs, but its over-expression results in ectopic DNA damage**

Since Gen is a nuclease involved in DNA damage repair, we measured the levels of DNA damage by determining the γH2Av index in diploid NBs, as before [33]. γH2Av recognizes double strand breaks, and it is an early marker of the DNA damage response. In Gen$^{mut}$ NBs, the γH2Av index was not increased, even if a few outliers could be easily recognized (Fig 3A, 3B). In strong GenOE brains, an increase in the γH2av index was noticed, suggesting that over-expressing Gen generates increased DNA damage levels.

We next determine if Gen activity is required in conditions of replicative stress- the slowing or stalling of replication fork progression [45], which leads to reduced replication fidelity and DNA double strand breaks. We used Aphidicolin (APH), a DNA polymerase inhibitor that causes replication fork stalling and therefore replicative stress [46]. We used APH at 50μM for 1h30 followed by immediate fixation or alternatively release for 30 min in the absence of APH (Fig 3C). We verified that these conditions resulted in EdU incorporation decrease (S4A, S4B Fig), attesting to an effect in DNA replication as expected. We chose a short incubation time, as the NB cell cycle is quite fast- around 1h25 min on average (Fig 2C) in Ctrl brains.

We scored the γH2Av index in interphase NBs and found an increase in Ctrl, Gen$^{mut}$ and GenOE APH-treated brains, when compared to DMSO treated brains (Fig 3D, 3E and 3G- note also that DMSO alone increases γH2Av signals in

   

**Fig 3. Analysis of the γH2av index in conditions of replicative stress does not show increased levels in Gen^mut NBs but GenOE NBs present high levels of DNA damage in a HR independent manner.** (A) Pictures of Ctrl, Gen^mut and GenOE brain lobes labeled with antibodies against γH2av (in pink). DNA is shown in cyan. (B) Dot plot graph showing the γH2av index of interphase NBs of the indicated genotypes. (B) Schematic diagram of the

experimental set up used to induce replicative stress. Brains were incubated in 50μM of APH for 1h30 and fixed or washed and subsequently released in APH-free medium for 30 min. (D) Pictures of Ctrl, Gen^mut and GenOE interphase NBs labelled with antibodies against γH2Av (pink in the merged panels and in grey on the right panels). The white dashed circles surround the nuclei. (E–J) Dot plot graphs showing the γH2Av index of the indicated genotypes in cells treated with APH for 1h30 (E,G. and I) or after release (F, H and J). Bars indicate the mean ± SEM. Statistical significance is shown and determined by Kuskal-Wallis non-parametric test. Experiments were repeated at least two times with a minimum of 10 brain lobes analyzed per condition.

certain conditions). Comparison of Ctrl and Gen^mut NBs showed similar γH2Av indices in APH, while GenOE NBs displayed a higher γH2Av index. These findings suggest that Gen is not essential for repairing DNA damage caused by replicative stress in diploid NBs, however increased Gen activity seems to result in ectopic DNA damage in conditions of replicative stress.

We then measured the γH2Av index of interphase NBs, after APH washout and recovery for 30 min- release. In Ctrl and Gen^mut brains, an impressive decrease in the average γH2Av index was seen in the samples previously treated with APH, now comparable to DMSO- treated brains (Fig 3D and 3F). In contrast, in GenOE brains, the γH2Av index remained higher than Ctrl brains (Fig 3D and 3H), even if decreased when compared to samples taken before the release. We concluded that in conditions of replicative stress, Gen is not essential for DNA damage repair in NBs and that increasing Gen levels results in higher levels of DNA damage. Further, repair during release does not follow the same dynamics as in Ctrl brains.

Gen is a resolvase involved in HJ resolution after nuclear envelope breakdown, downstream of HR occurring during S/G2 phases [18,20,47]. To test if GenOE recruitment and consequent ectopic DNA damage in APH treated conditions was solely dependent on HR, we depleted Rad51 from diploid NBs. We reasoned that if the only function of *Drosophila* Gen nuclease is related with HJ resolution, in conditions where HR cannot occur, an increase in Gen activity- provided by GenOE- should not result in increased γH2Av index. In flies, it has been reported that Rad51-dependent HR is not essential for DNA repair, except if exposed to damaging factors such as APH [48]. Indeed, an increase in γH2Av index was detected in Rad51^RNAi brains after APH treatment, even if not as important as in Ctrl diploid brains (Fig 3I). Interestingly, the combination of Rad51^RNAi and GenOE brains treated with APH showed increased DNA damage levels, but these were comparable with γH2Av levels found in GenOE brains, even after release (Fig 3J). These results indicate that Gen can still be recruited and induce ectopic DNA damage in conditions where HR and thus HJ resolution are compromised. Altogether, our data shows that GenOE induces ectopic DNA damage in conditions of replicative stress and that its recruitment can occur independently of HR.

### Gen over-expression promotes unscheduled polyploid cell proliferation in a nuclease activity independent manner

Since GenOE accelerated the cell cycle timing in diploid NBs, we next tested the consequences of Gen over-expression in polyploid NBs. Surprisingly, polyploid, strong GenOE brains contained larger NBs than the ones found in polyploid brains. The mild GenOE condition also resulted in increased polyploid NB area, but not to the same extent as strong GenOE (Fig 4A, 4B). To determine the role of nuclease activity in promoting polyploid cell proliferation, we generated a catalytic nuclease-dead Gen transgene by replacing the E-G-V-A residues in the first XPG domain and the EAEA in the second XPG domain by alanine residues, and placed it under the UAS-promoter, fused to mCherry (Fig 4C). We refer to this construct as GenND. In this case, analysis of endogenous Gen protein was still present, as we failed, to obtain the combination of GenND in a Gen^mut background. In diploid brains, GenND behaved similarly to GenOE, remaining mainly associated with the cytoplasm throughout interphase and no obvious morphological defects were noticed (S4C Fig). In these brains, the mitotic index was slightly increased when compared to control brains, but there were no obvious mitotic defects like lagging chromosomes or chromosomal bridges (S4D Fig). Further, analysis of DNA damage did not reveal increased

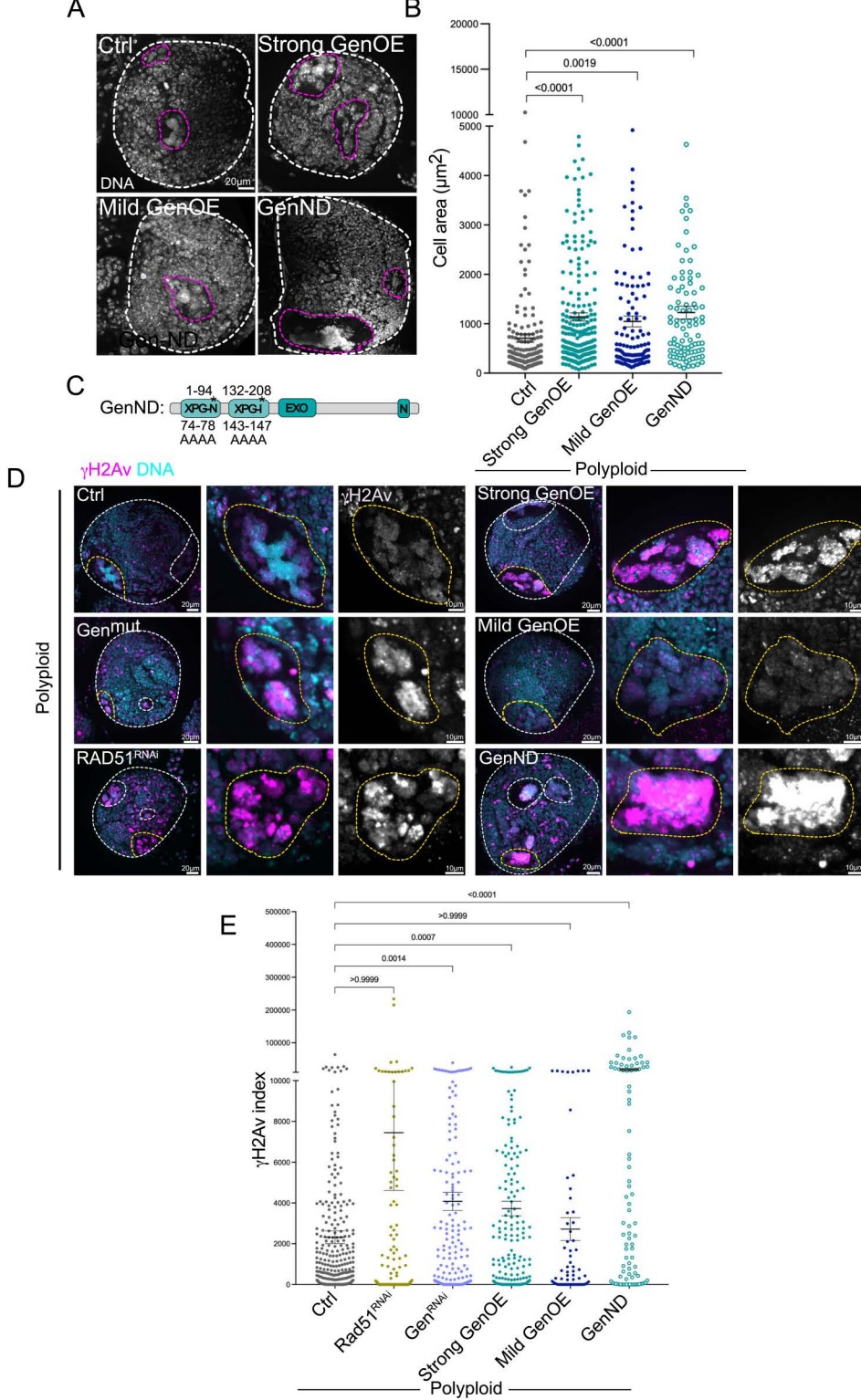

**Fig 4. Increased Gen levels favor polyploid cell proliferation independently of DNA damage levels.** (A) Pictures of polyploid brains (Ctrl) and other combinations of the indicated genotypes showing DNA labeling (grey). The white dashed line surrounds the brain lobe and the pink dashed lines surround polyploid NBs. (B) Dot plot showing cell area of the indicated genotypes. (C) Diagram of the Gen nuclease dead (ND) transgene. Numbers

on the top indicate the position of the domains and on the bottom the residues replaced by Alanines. This construct was fused at the C-terminus with a mCherry (mCh) tag. (D) Pictures of polyploid brains (Ctrl) and other combinations of the indicated genotypes labelled with γH2av antibodies (pink in the merged panel and grey in the inset panels on the right). DNA is shown in cyan. The white dashed lines surround brain lobes and polyploid NBs, while the yellow circles surround the NBs magnified in the insets. (E) Dot plot graphs showing the γH2av index of the indicated genotypes. In B and E, bars show the mean±SEM. Statistical significance is shown and determined by Kuskal–Wallis non-parametric test. Experiments were repeated at least two times with a minimum of 15 brain lobes analyzed per condition.

in DNA damage in diploid NBs expressing GenND (S4E Fig). In polyploid NBs, GenND localized similarly to GenOE (S4F Fig). Interestingly, polyploid, GenND NBs displayed increased cell area when compared to polyploid NBs, with a mean even slightly higher than polyploid, GenOE (Fig 4A, 4B).

The differences found between polyploid, Rad51$^{RNAi}$/polyploid Gen loss-of-function conditions and polyploid, Gen over-expressing conditions were striking. We predicted that defects in DNA repair may account to delay cell cycle progression in unscheduled polyploid cells. Indeed, the γH2av index of both polyploid,Rad51$^{RNAi}$ or polyploid, Gen$^{RNAi}$ NBs were higher than in Ctrl polyploid cells (Fig 4D, 4E). Surprisingly however, polyploid, strong GenOE and polyploid, GenND NBs displayed high γH2av indices with the latter condition showing a massive increase. In contrast, polyploid, mild GenOE did not show increased γH2av index (Fig 4D, 4E). These results suggest that Gen's nuclease activity is required to repair the typical DNA damage found in non-physiological polyploid cells [10] and that this repair requires, at least in part- HR. Further, they also suggest that a novel function of this protein seems to be implicated in promoting unscheduled polyploid cell proliferation independently of DNA damage repair.

## Gen levels influence cell cycle progression in polyploid NBs through DNA replication

We wanted to further test Gen's role in DNA damage repair and in the capacity to sustain or influence polyploid cell proliferation. Considering the differences in cell cycle timing in diploid Gen$^{mut}$ and GenOE brains described above, we hypothesized that Gen may also influence cell cycle progression in polyploid cells. Unfortunately, multiple attempts to film polyploid, Gen$^{mut}$ or polyploid, GenOE brains for extended periods of time were unfruitful, as these brains were quite fragile. We therefore took advantage of a characteristic of Polyploid NBs- cell cycle asynchrony- where the multiple nuclei of a given polyploid cell do not transition into mitosis in a synchronous manner. This is also the case of mouse neural stem cells and cancer cells containing multiple nuclei [33]. In addition, such asynchrony is a source of DNA damage, as the nuclei that are not ready to undergo chromosome segregation can fragment upon forced mitotic entry imposed by the neighboring nuclei within the polyploid cell [33]. We reasoned that if Gen has a role in influencing the proliferation capacity of polyploid cells by influencing cell cycle progression, this should also impact the frequency of cell cycle asynchrony. Using PH3 as a marker to identify mitotic nuclei, we quantified the frequency of asynchrony by considering whether in multinucleated cells in mitosis (PH3+), one or more nuclei were PH3- (Fig 5A). The asynchrony frequency was extremely reduced in polyploid, Gen$^{mut}$, while it was increased in polyploid, GenOE brains (Fig 5B). Moreover, the mitotic γH2Av indices were also reduced and increased respectively (Fig 5C–5E), indicating that delaying or accelerating cell cycle progression impacts the levels of DNA damage observed during mitosis.

Considering these results, we next asked how Gen levels can influence cell cycle asynchrony in polyploid NBs. If cells with multiple nuclei -typical of unscheduled polyploidy generated by repeated cytokinesis failure- entered mitosis with un-replicated DNA, the asynchrony may be established in S-Phase where uneven DNA replication may account for cell cycle asynchrony. We tested this possibility by performing 2h EdU pulses in polyploid Ctrls, polyploid, Gen$^{RNAi}$ and polyploid, GenOE brains. The percentage of EdU coverage was calculated by dividing the area covered by EdU by the total nuclear area. In all conditions, a continuum of EdU coverages was seen, with certain nuclei showing a complete absence of EdU (not replicating during the 2h pulses), or almost 100% coverage, showing that EdU had been incorporated throughout the entire nuclear area. Importantly, the percentage of EdU coverage was reduced in polyploid, Gen$^{RNAi}$ NBs

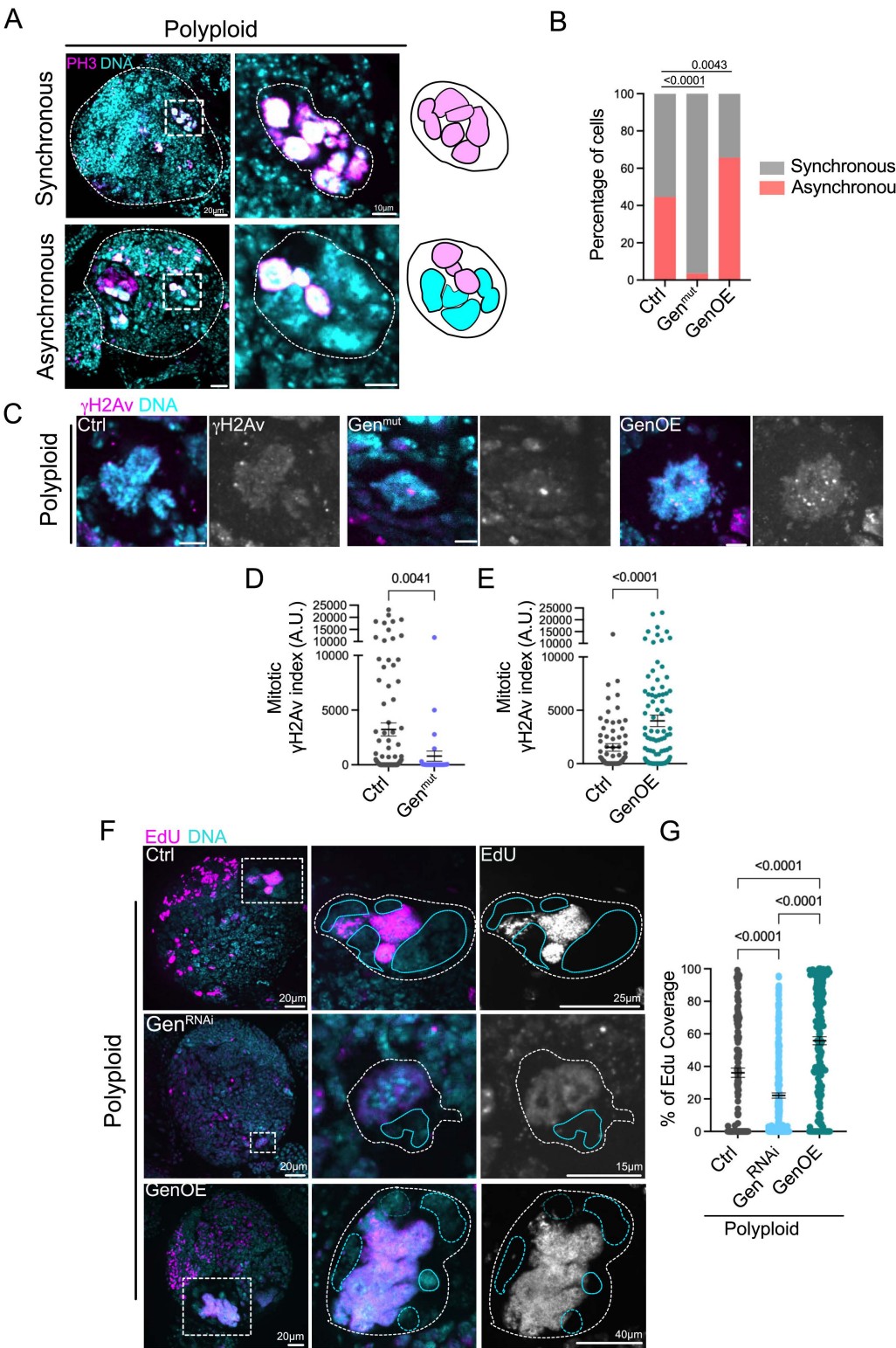

**Fig 5. Polyploid, Gen^mut and Polyploid, GenOE show opposite outcomes in terms of cell cycle progression and EdU incorporation.** (A) Pictures of Polyploid brains brain lobes labelled with antibodies against PH3 (pink). DNA is shown in cyan. An example of a large polyploid cell showing multinu-cleated polyploid NBs undergoing mitosis in a synchronous (top) or asynchronous (bottom) manner. On the right, schematic diagrams to illustrate the

synchrony/asynchrony. (B) Graph bar showing the percentage of asynchronous polyploid NBs in the indicated genotypes. (C) Pictures of mitotic NBs of the indicated genotypes labelled with antibodies against γH2Av (pink and grey panels on the right). DNA is shown in cyan. (D, E) Dot plot graphs showing the γH2av index of mitotic NBs. Cells were chosen in advanced mitotic stages (but before anaphase), so that all nuclei of a given cell has condensed their chromosomes and progressed through mitosis. (F) Pictures of Polyploid brain lobes of the indicated genotype after a 2h EdU pulse. EdU is shown in pink and grey on the right panels and DNA in cyan. NB insets are highlighted with white dashed squares. The blue lines surround EdU- nuclei. (G) Dot plot graphs showing the Edu signal coverage of the indicated genotypes. In D-E and G, bars show the mean ± SEM. Statistical significance is shown and determined by Kuskal-Wallis non-parametric test. Experiments were repeated at least two times with a minimum of 15 brain lobes analyzed per condition.

compared to Ctrl polyploid cells (Fig 5F, 5G), suggesting that DNA replication was blocked in these cells. In contrast, polyploid GenOE displayed increased EdU coverage, with multiple cells reaching close to 100% (Fig 5F, 5G). Taken together, these findings suggest that Gen is absolutely required for DNA replication in polyploid conditions and that the levels of Gen influence cell cycle progression and synchrony towards mitotic entry.

## Discussion

In this study we explored the function of Gen nuclease *in vivo* using a variety of genetic conditions. Our results reveal three important aspects concerning Gen nuclease biology.

First, in diploid conditions Gen loss-of-function or over-expression are compatible with development in flies. Mitotic errors were observed in both conditions and interestingly Gen^mut and GenOE impacted in opposite ways mitotic timing and cell cycle progression. Combining fixed analysis with time-lapse microscopy showed that in Gen^mut, the increase in mitotic index is explained by a longer time spent in mitosis. Extended mitosis is most likely due to the maintenance of the spindle assembly checkpoint in an active status. In sharp contrast, the faster mitotic timing, typical of GenOE seems to be explained by more cells entering mitosis at a given time than in Ctrl brains but then going faster through mitosis, explaining also the faster cell cycles seen in these NBs. Why over-expressing GenOE results in faster cell cycles and faster mitosis, is difficult to explain, but it may be related with a possible function in DNA damage signaling, initially proposed in *C. elegans* [49]. The defects in chromosome segregation, such as chromosome bridges and acentric chromosomes may be explained by inaccurate DNA damage repair generating a decrease in DNA damage signaling (lower γH2aV index). More difficult to explain is the presence of lagging chromosomes found in both Gen^mut and GenOE diploid NBs. These may reflect defects in centromeric structure impacting kinetochore assembly. Interestingly, analysis of GenND in diploid NBs revealed no mitotic defects or increase in DNA damage signaling. This shows that the defects observed in GenOE diploid NBs result from nuclease activity.

Second, still considering diploid conditions, our study reveals that Gen loss-of-function does not perturb DNA damage repair in NBs, even after inducing replicative stress through APH treatment. Increased Gen levels result in ectopic DNA damage that is not repaired using the same dynamics as Ctrl NBs after drug release. The origins of GenOE ectopic DNA damage remain to be identified. The persistence of DNA damage in conditions where HR cannot occur suggest that Gen recruitment, activity and capacity to generate DNA damage are at least partly independent of HR. The lack of differences between Gen^mut and Ctrl NBs in APH treated conditions, suggests that other mechanisms are involved in DNA damage repair in this cell type. It is important to mention that in conditions of high DNA damage, defects in *Drosophila* brain development are easy to identify. This is the case of mutations in Fen1, another endonuclease, where high γH2av levels were described, together with increased necrosis and severe brain size reduction [50]. These results indeed suggest that other factors may compensate for lack of Gen, like Mus81, as described in yeast and B lymphocytes [25–27,51].

Finally, we report here a very unexpected requirement for Gen in unscheduled polyploid cell proliferation. Polypoid, Gen^mut NBs show reduced cell size explained by low proliferation levels. Further, they also show reduced cell cycle asynchrony and low DNA damage levels in polyploid mitotic cells, suggesting delayed cell cycle progression allowing mitotic entry in a more synchronous manner as previously described for CDK1 inhibition [33]. Similarly, Polyploid, Rad51^RNAi NBs

also display important cell size reduction. At first glance, it is tempting to assume that a reduction in proliferation results from increased DNA damage due to lack of efficient HR and HJ resolution. Certainly, this may be the case in respect to the role of Rad51 in HR, which is essential to repair DNA damage generated during S-Phase. However, our data demonstrate that depletion of other members of the HJ resolution pathways does not affect polyploid cell size. Additionally, we found that polyploid, GenOE or polyploid, GenND show concomitantly increased cell size and result in high levels of DNA damage- even higher in GenND conditions. Therefore, polyploid cell proliferation does not seem to be restrained by high DNA damage levels, at least generated in GenOE or GenND dependent manner. Moreover, these results also suggest that the proliferative advantage described here is independent of Gen's nuclease activity. These data strongly suggest that there is another role for this protein, probably independent of DNA repair, which seems to be related with DNA replication during S-Phase.

Interestingly, a role for *Drosophila* Gen in lagging strand replication fork cleavage has been reported which is dependent on Gen's enzymatic activity [29]. How does Gen's activity influence DNA replication rates is not known, but since polyploid, GenND NBs are also larger than polyploid NBs, it is possible that this function does not depend on nuclease activity. In *C. elegans,* the Gen orthologue shows distinct functions in DNA damage signaling and repair [49]. In light of our results it is tempting to propose that other functions non-related with DNA damage repair have to be attributed to this enzyme.

Intriguingly, this novel role for Gen nuclease seems to be more obvious in polyploid than in diploid conditions. The reason for such difference remains to be explored in future work, but it is conceivable that the unbalanced proteomes typical of non-physiological polyploid cells may buffer the redundancy that compensates certain insults (like DNA damage or replicative stress) in diploid conditions. Altogether, our work suggests that Gen inhibition may be interesting to consider from a therapeutically point of view: it does not extensively affect diploid cells, but severely inhibits unscheduled polyploid cell proliferation. Exploring the cell cycle role of Gen during S-Phase in unscheduled polyploid cells is an exciting avenue for upcoming studies.

## Materials and methods

### Fly husbandry

Flies were raised on cornmeal medium (0.75% agar, 3.5% organic wheat flour, 5.0% yeast, 5.5% sugar, 2.5% nipagin, 1.0% penicillin-streptomycin and 0.4% propionic acid). Fly stocks were maintained at 18°C. Crosses were carried out in plastic vials and maintained at 25°C unless differently specified. Stocks were maintained using balancer inverted chromosomes to prevent recombination. In all experiments, larvae were staged to obtain comparable stages of development. Egg collection was performed at 25 °C for 24h. After 5 days of development at 25 °C, third instar larvae were used for brain dissection.

### Fly stocks

Control flies used in this study vary depending on the experiment. For experiments performed in diploid brain lobes, controls were *wild-type* (WT) flies $w^{1118}$ (BL 3605), or flies carrying the Worniu-Gal4 driver and/or the insertion site without the UAS transgene (for experiments conducted in UAS-Gen-mCh, UAS-GenND, UAS-Mre11-mCh and all the UAS-RNAi lines). For Gen$^{mut}$ analysis, the $Gen^{z-5997}$ allele, carrying a frameshift mutation at codons 374–5 in its conserved nuclease domain [52] was combined with a deficiency- Df(3L)Exel6103- which is essential to uncover the hypersensitivity of the $Gen^{z-5997}$ allele [39] For experiments performed in polyploid brain lobes, controls were $sqh^1$ [35]or $sqh^1$, Worniu-Gal4. See Table 1 for a list of all *Drosophila* stocks.

### Generation of transgenic flies

PhiC31 integrase-mediated transgenesis was used to insert UAS-Gen-mCh, UAS-GenND-mCh, Ubq-Gen-NeG and UAS-Mre11-mCh transgenes in a precise site of the *Drosophila* genome. The transgene was cloned in a P[acman] plasmid

**Table 1. List of *Drosophila* Stocks used in this study.**

| Stock name | Source | Reference |
|---|---|---|
| *w*[1118] | BL3605 | Used in [10,33,34] |
| y[1] w[1118]; PBac{y[+]-attP-3B}VK00037. 2nd chromosome attP docking site for phiC31 integrase-mediated transformation | BL 9752 | |
| *Sqh¹* | Karess lab | [35] |
| Genz5997 | Sekelsky lab | [39] |
| Df (3L) Exe 16103 | BL7582 | [39] |
| Gen[RNAi] | BL42787 | [53] |
| SIX1[RNAi] | BL34949 | [54] |
| Mus81[RNAi] | BL65012 | [53] |
| RAD51[RNAi] | BL51926 | [53] |
| UAS-Gen-mCh | This study | |
| UAS-Mre11 | This study | |
| UAS-GenND | This study | |
| Ubq-Gen-NeG | This study | |
| H2Av-mRFP | BL23651 | [55] |
| Worniu-Gal4 | BL56553- a gift from the Doe lab | Used by in [10,33,34] |

containing the attB site (from bacterial genome) and injections were performed by BestGene (BestGene Inc, CA, USA). The recipient *Drosophila* stock for injection (y[1] w[1118]; PBac{y[+]-attP-3B}VK00037), containing attP sites (from the PhiC31 bacteriophage genome), was obtained from Bloomington Drosophila Stock Center (BL 9752), Indiana University, IN, USA. Integration was targeted to Chromosome II at chromosomal location 2L:1582820.

## Molecular biology

*Drosophila* genomic DNA was extracted from 40 *wild type* (*w1118,* BL3605) adult flies using the following protocol: flies were homogenized on ice in 250 μL Tris HCl 0.1 M pH 9, EDTA SDS 1%, and incubated for 30 min at 70°C. 35 μL KAc 8M pH 7.8 was added and incubated on ice for 30 min, centrifuged for 15min and 250 μL phenol-chloroform was added to the supernatant. Samples were centrifuged for 5min and 150μL isopropanol was added to the supernatant and centrifuged for 5min to precipitate DNA. The DNA pellet was washed with 70% EtOH, centrifuged for 5min, dried and resuspended in Tris EDTA.

For amplification of *Drosophila* Gen and Mre11, 30 cycles of PCR were performed in 50 μL volume using 100 ng purified genomic DNA as a template, 1 μL dNTP mix (10mM), 1 μL of each primer nucleotides (25mM). Annealing was performed for 30 seconds between 55–65°C depending on the sequence to be amplified, and elongation at 72°C for 30 seconds/kb of the template length. Each PCR product was analyzed by agarose gel electrophoresis and purified from an agarose gel slice using a gel purification kit (28704, QIAGEN). Primers used for PCR and cloning of Gen are CGCGGATATCATGGG-CGTCAAGGAATTATGGGG and CGCGACTAGTATCACTAATCACTACCAGGTCATCC. Primers used for PCR and cloning of Mre11 are CGCGGATATCATGAATGGCACCACGACAGCAGAGC and CGCGACTAGTATCGGAATCATCCGA.

## Molecular cloning

1μg of Gen and Mre11 fragments amplified from *Drosophila* genomic DNA and carrying restriction sites for EcoRV and SpeI were digested for 1h at 37°C with 5U EcorV and 5U of Spe1, with 5 μL Cutsmart Buffer (New England Biolabs) in 50 μL total volume. pBlueScript II SK (+) (PBSK+) vector, containing an AscI restriction site at the 5' of a UAS-mCherry (Gen-Script) and a NotI restriction site at the 3' was also digested using the same protocol. Digestion products were purified using a PCR purification kit (QIAGEN 28104). Each fragment was inserted in PBSK-UAS-mCh using a 1:3 ratio of

vector: insert diluted in a total volume of 2,5 µl of TE (10 mM Tris-HCI, pH 8.0, 1 mM EDTA) buffer, and 2,5 µL of ligation mix (Takara, 6023), for 30 min at 16 °C. The ligation mixture was directly used for transformation with 50 µl *E. coli* competent cells (Dh5α, New England Biolabs, C2987) and several positive colonies were collected and purified by Miniprep (17106, QIAGEN). Constructs were first verified using restriction profile, and one positive clone was chosen to be sent to Eurofins Genomics for Sanger sequencing. Then, UAS-Gen-mCh and UAS-Mre11-mCh, were digested with AscI and NotI restriction enzymes (New England Biolabs) for 1 hour at 37°C, purified and ligated with a P[acman] plasmid for 30 min at 16°C. The ligation product was used to transform TransforMax EPI300 E. coli (C300C105, Epicentre). P[acman] was purified from several clones and length of the insert was tested by restriction enzyme digestion with AscI and NotI enzymes. One clone was used to generate transgenic flies by PhiC31 integrase-mediated transgenesis as before.

## Whole mount tissue preparation and imaging of *Drosophila* brains

Brains from third instar larvae were dissected in PBS and fixed for 30 min in 4% paraformaldehyde in PBS. They were washed 3 times in PBST 0.3% (PBS, 0.3% Triton X-100 (Sigma T9284), 10 min for each wash) and incubated for several hours in agitation at room temperature and overnight at 4°C with primary antibodies at the appropriate dilution in PBST 0.3% (a list of antibodies used in this study is found in Table 2). Tissues were washed three times in PBST 0.3% (10 min for each wash) and incubated overnight at 4°C with secondary antibodies diluted in PBST 0.3%. Brains were then washed 2 times in PBST 0.3% (30 min for each wash), rinsed in PBS and incubated with 3µg ml−1 DAPI (4′,6-diamidino-2-phenylindole; Sigma Aldrich D8417) at room temperature for 30 min. Tissues were then washed in PBS at room temperature for 30 min and mounted on mounting media. A standard mounting medium was prepared with 1.25% n-propyl gallate (Sigma P3130), 75.0% glycerol (bidistilled, 99.5%, VWR 24388–295), 23.75% H2O). Images were acquired with 40x oil objective (NA 1.4) on two a wide-field Inverted SpinningDisk Confocal Gattaca/Nikon (a Yokagawa CSU-W1 spinning head mounted on a Nikon Ti-E inverted microscope equipped with a camera complementary metal-oxide semiconductor 1,200 × 1,200 Prime 95B; Photometrics), controlled by Metamorph software. Interval for z-stacks acquisitions was set up from 0.5µm to 1µm.

## Live imaging of *Drosophila* brains

Mid third-instar larval (L3) brains expressing fluorescent proteins were dissected in Schneider's *Drosophila* medium supplemented with 10% heat-inactivated fetal bovine serum, penicillin (100 U/ml), and streptomycin (100µg/ml) (Penicillin-Streptomycin 15140, Gibco). Several brains were placed on a glass-bottom 35 mm dish (P35G-1.5-14-C, Mat-Tek Corporation) with approximately 10µL of medium, covered with a permeable membrane (Standard YSI), and sealed around the membrane borders with oil 10S Voltalef (VWR BDH Prolabo). Images were acquired with 60x oil objective (NA 1.4) on two a wide-field Inverted SpinningDisk Confocal Gattaca/Nikon (a Yokagawa CSU-W1 spinning head mounted on a Nikon Ti-E inverted microscope equipped with a camera complementary metal-oxide semiconductor 1,200 × 1,200

**Table 2. List of antibodies used in this study.**

| Antibody | Dilution | Source | Identifier |
|---|---|---|---|
| Rabbit polyclonal anti γH2Av | 1:500 | Rockland immunochemicals | 600-401-914 |
| Mouse monoclonal anti-Phospho histoneH3 (Ser10) | 1:500 | Cell Signalling Technologies | CST 9706 |
| Guinea Pig anti Deadpan | 1:1000 | Our lab | [42] |
| Mouse monoclonal anti Neon green | 1:500 | Chromotek | 32F6 |
| mCherry | 1:500 | abcam | Ab213511 |
| Goat anti rabbit Alexa 488 | 1:250 | Thermofisher scientific | A11008 |
| Goat anti mouse Alexa 546 | 1:250 | Thermofisher scientific | A11030 |
| Goat anti guinea pig Alexa 647 | 1:250 | Thermofisher scientific | A11073 |
| Phalloidin Alexa 647 | 1:250 | Thermofisher scientific | A22287 |

Prime 95B; Photometrics), controlled by Metamorph software. Images were acquired at time intervals spanning from 1min (diploid conditions) to 10min (polyploid conditions) and 30–50 Z-stacks of 1–1.5μm.

### EdU incorporation in polyploid cells

3rd instar larval brains were dissected in Schneider's *Drosophila* medium and incubated for 2 hours at 25°C in the same medium with 100mM EdU. Brains were then washed in PBS, fixed and immunostained. EdU detection was performed after secondary antibody detection, according to the manufacturer instructions (C10640, Molecular Probes, TermoFisher Scientific, Waltham, MA, USA). Image acquisition and treatment were made as described for whole mount preparation of tissues. Quantitative analysis of EdU nuclear coverage was performed as described below.

### Aphidicolin treatment

Brains were dissected as before and incubated in DMSO anhydrous (5.89569 Sigma) or Aphidicolin (50μM) (A0781 from Sigma) diluted in Schneider's medium for exactly 1.30h. After this period, they were either fixed immediately, or washed 3x in Schneider's *Drosophila* medium followed by incubation in the same medium for 30min.

### Quantitative analysis of DNA damage

Image analysis was performed using Fiji. Images were imported as Z-stack of 4 channels (PH3-red, γH2Av – green, DAPI – blue, Phalloidin – far red). Individual cells were manually cropped from the original image, as Z-stacks of 5 Z each. For DNA damage and cell area analysis, a Fiji macro was developed by Anne-Sophie Mace, (Institut Curie, UMR144) to automate the signal quantification. For each image containing an individualized cell, cell area was manually segmented from a projection of 5 Z stacks on the far-red channel using the freehand selection tool. Similarly, nuclei were manually segmented from a projection of the 5 Z stacks ion the blue channel, using the freehand selection tool. These two segmentations were automatically saved as regions of interest (ROIs), and the green channel was used to separate positive pixels from negative pixels by a thresholding operation. We assigned a constant threshold value of 300 for DNA damage quantification, and 1000 for detecting PH3- nuclei. The macro generates an output for each cropped images, indicating for each image: cell area in pixel2, nuclear area in pixel2, the average intensity of the green channel in the ROI (named DNA damage fluorescence intensity), the area of positive pixels of the green channel in the ROI (named DNA damage area), the area of positive pixels of the red channel in the ROI (name mito2c area). In addition, the macro generates a montage, which is a RGB image providing an overview of the measured ROIs. Once the output has been obtained, cell area was adjusted in μm2, based on the pixel value given by the microscope and objective. To obtain the γH2Av index, DNA damage area was divided by the nuclear area, which provides the DNA damage coverage. DNA damage coverage was expressed as a percentage of coverage, and multiplied by the DNA damage fluorescence intensity, to give the γH2Av index in arbitrary units (A.U).

### Quantitative analysis of mitotic index

For mitotic index quantification, images of brain lobes were imported as Z-stacks of 4 channels (DPN red, γH2Av – green, DAPI – blue, Ph3 – far red). DPN+ (green) NBs were manually counted using Fiji Software, to obtain the total number of NBs per lobe. Among these, PH3+NBs were manually counted for each lobe. The number of PH3+ (DPN+) NBs was divided by the total number of NBs (DPN+) and expressed as a percentage.

### Image processing and statistical analysis

Image processing used Fiji software and figures were mounted using Affinity Designer. Statistical analysis was performed with GraphPad Prism (RRID SCR 002798) version 9.00 for Mac (GraphPad So[ware), using the tests mentioned in the figure legends.

## Supporting information

**S1 Fig. Analysis of Polyploid, Rad51 brains reveals a mitotic arrest.** (A) Pictures of Ctrl polyploid and Polyploid, Rad51^RNAi labelled with antibodies against PH3. PH3 and Actin are showed in grey. White dashed lines surround the brain lobes and pink dashed lines surround mitotic cells. (B, C) Dot plot graph showing the mitotic index (B) and the mitotic γH2av index (C) of the indicated genotypes. Bars indicate the mean ± SEM. Statistical significance is shown and determined by a Two-tailed Mann-Whitney test. Experiments were repeated at least two times with a minimum of 15 brain lobes analyzed per condition.
(EPS)

**S2 Fig. Gen^mut diploid NBs show delayed mitosis and mitotic errors.** (A) Pictures of Ctrl (left) and Gen^mut (right) diploid brain lobes labelled with antibodies against the neural stem cell marker- DPN (pink). DNA is shown in cyan. (B, C) Dot plot graphs showing the number of NBs per brain lobe (B) and the mitotic index (C) of the indicated genotypes. (D) Stills of time lapse movies of Ctrl WT diploid and Gen^mut NBs expressing H2B-RFP to visualize chromosomes during mitosis. The pink dashed lines surround NBs. Time is in min and time zero was define at nuclear envelope breakdown. (E) Top- Pictures of NBs in anaphase illustrating no errors, lagging chromosomes, where a v-shaped chromatid can be distinguished oriented towards one pole, acentric where two pieces of chromatin without a visible centromeric constriction are positioned on opposite chromosome poles and DNA bridges, where thin DNA structures link both anaphase poles. Bottom- schemes illustrating the different chromosome behaviors. (F) Graph bar showing the percentage of cells in each category in Ctrl and Gen^mut NBs. In B and C, bars indicate the mean and SEM. Statistical significance is shown and determined by Two-tailed Mann-Whitney tests. Each dot corresponds to one brain lobe. Experiments were repeated at least two times with a minimum of 10 brain lobes analyzed per condition.
(EPS)

**S3 Fig. Characterization of Strong GenOE and mild GenOE diploid brains.** (A) Schematic diagram of the mild GenOE transgene, where the full coding region was fused at the C-terminus with a Neon green (NeG) tag. (B) Left- brain lobe expressing the mild GenOE transgene (green). DNA is in blue. Right- insets of interphase (I) and mitotic (M) NBs. Below mild GenOE is shown in grey. The numbered squares correspond to the inset numbers. Note that interphase NBs can show very low nuclear Gen levels (1) or higher Gen levels, while mitotic NBs show Gen spreading throughout the cytoplasm overlapping with chromosomes. (C) Dot plot graph showing the mitotic indices of the indicated genotypes. (D-G) Dot plot graphs showing the NB number (D), the mitotic index (E), mitotic duration (F) and cell cycle duration of the indicated genotypes. (H) Stills of time lapse movies of diploid NBs expressing Mre-11mCherry (top) and H2B-GFP (bottom) to visualize chromosomes. The pink and blue dashed lines surround NBs and the nucleus respectively. Time is in min. Time zero was defined at nuclear envelope breakdown. (I) Dot plot graph showing the mitotic index of the indicated genotypes. (J) Graph bar showing the percentage of anaphase in each category in Ctrl and strong GenOE NBs. For anaphase errors- refer to the schemes and pictures of S2E Fig. For C–G and I, bars show the mean ± SEM. Statistical significance is shown and determined by Two-tailed Mann-Whitney tests. Experiments were repeated at least two times with a minimum of 10 brain lobes analyzed per condition.
(EPS)

**S4 Fig. Conditions of replicative stress inhibit DNA replication and GenND characterization.** (A) Pictures of WT diploid brains lobes treated with DMSO (left) or APH (right) labelled with EdU (pink and grey in the bottom panels). DNA in shown in cyan. (B) Dot plot graph showing the percentage of EdU positive diploid NBs after the indicated treatments. Experiments were repeated at least two times with a minimum of 10 brain lobes analyzed per condition. (C) Pictures of diploid brains expressing UAS-GenND-mCh (shown in pink on the left and in grey on the right). The white dashed line delimits brain lobes. (D) Dot plot graph showing the mitotic index of diploid Ctrl and GenND brains. (E) Dot plot graph

showing the γH2av index of diploid Ctrl and GenND brains. (F) Pictures of Polyploid, strong GenOE (left) and Polyploid, GenND (right) brain lobes. In pink Gen and DNA in cyan. White dashed lines surround brain lobes and polyploid NBs. Bars show the mean±SEM. Statistical significance is shown and determined by Two-tailed Mann-Whitney tests. (EPS)

**S1 Data. An Excel document containing all the raw data is available.** Legend for Excel spreadsheets containing all the Raw data from Budzyk et al, 2025. Spreadsheet 1- Raw data related with Fig 1D. Spreadsheet 2- Raw data related with Fig 1G. Spreadsheet 3- Raw data related with Fig 2B, 2C and 2F. Spreadsheet 4- Raw data related with Fig 2I, J. Spreadsheet 5- Raw data related with Fig 3B and 3E–J. Spreadsheet 6- Raw data related with Fig 4B. Spreadsheet 7- Raw data related with Fig 4E. Spreadsheet 8- Raw data related with Fig 5B, 5D, E. Spreadsheet 9- Raw data related with S1B, C Fig. Spreadsheet 10- Raw data related with S2B, C and S2F Fig. Spreadsheet 11- Raw data related with S3C–G Fig and S3I Fig. Spreadsheet 12- Raw data related with S4B and S4D, E Fig. (XLSX)

## Acknowledgments

We thank the PiCT-IBiSA platform and Nikon Imaging Center at the Institut Curie and Vincent Fraisier for excellent and continuous support on image acquisition. We thank S. Gemble, V. Marthiens, A. Terrizzano and R. Salame for helpful discussions and comments on the manuscript, and L. Jawish for help with dissections and immunostaining. We thank J. Sekelsky (University of North Carolina, NC, USA) for reagents and G. Mazon (IGR, Villejuif, FR), S. West (CRICK Institute, London UK) and J. Matos (Vienna University, Vienna, AT) for insightful discussions.

## Author contributions

**Conceptualization:** Manon Budzyk, Renata Basto.

**Formal analysis:** Manon Budzyk, Anthony Simon, Renata Basto.

**Funding acquisition:** Renata Basto.

**Investigation:** Manon Budzyk, Anthony Simon, Renata Basto.

**Methodology:** Anne-Sophie Mace.

**Project administration:** Renata Basto.

**Supervision:** Renata Basto.

**Validation:** Manon Budzyk, Anthony Simon, Renata Basto.

**Writing – original draft:** Renata Basto.

**Writing – review & editing:** Renata Basto.

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
