## [Decision Letter · Decision Letter 0]

19 Mar 2025

PGENETICS-D-25-00127

A novel DNA repair-independent role for Gen nuclease in promoting unscheduled polyploidy cell proliferation

PLOS Genetics

Dear Dr. Basto,

Thank you for submitting your manuscript to PLOS Genetics. After careful consideration, we feel that it has merit but does not fully meet PLOS Genetics's publication criteria as it currently stands. Therefore, we invite you to submit a revised version of the manuscript that addresses the points raised during the review process.

Please submit your revised manuscript within 60 days. If you will need more time than this to complete your revisions, please reply to this message or contact the journal office at plosgenetics@plos.org. Please include the following items when submitting your revised manuscript:

We look forward to receiving your revised manuscript.

Kind regards,

louise cheng

Academic Editor

PLOS Genetics

Fengwei Yu

Section Editor

PLOS Genetics

Aimée Dudley

Editor-in-Chief

PLOS Genetics

Anne Goriely

Editor-in-Chief

PLOS Genetics

**Journal Requirements:**

At this stage, the following Authors/Authors require contributions: Manon Budzyk, Anthony Simon, Anne-Sophie Mace, and Renata Basto. Please ensure that the full contributions of each author are acknowledged in the "Add/Edit/Remove Authors" section of our submission form.

The list of CRediT author contributions may be found here: https://journals.plos.org/plosgenetics/s/authorship#loc-author-contributions

https://journals.plos.org/plosgenetics/s/submission-guidelines#loc-parts-of-a-submission

3) We noticed that you used the phrase 'not shown' in the manuscript. We do not allow these references, as the PLOS data access policy requires that all data be either published with the manuscript or made available in a publicly accessible database. Please amend the supplementary material to include the referenced data or remove the references.

Potential Copyright Issues:

i) Figures 1A, 1C, and S2E. Please confirm whether you drew the images / clip-art within the figure panels by hand. If you did not draw the images, please provide (a) a link to the source of the images or icons and their license / terms of use; or (b) written permission from the copyright holder to publish the images or icons under our CC BY 4.0 license. Alternatively, you may replace the images with open source alternatives. See these open source resources you may use to replace images / clip-art:

6) When completing the data availability statement of the submission form, you indicated that you will make your data available on acceptance. We strongly recommend all authors decide on a data sharing plan before acceptance, as the process can be lengthy and hold up publication timelines. Please note that, though access restrictions are acceptable now, your entire data will need to be made freely accessible if your manuscript is accepted for publication. This policy applies to all data except where public deposition would breach compliance with the protocol approved by your research ethics board. If you are unable to adhere to our open data policy, please kindly revise your statement to explain your reasoning and we will seek the editor's input on an exemption. Please be assured that, once you have provided your new statement, the assessment of your exemption will not hold up the peer review process.

7) Please amend your detailed Financial Disclosure statement. This is published with the article. It must therefore be completed in full sentences and contain the exact wording you wish to be published.

3) If any authors received a salary from any of your funders, please state which authors and which funders.

Note: Please ensure that the funders and grant numbers match between the Financial Disclosure field and the Funding Information tab in your submission form. The funders must be provided in the same order in both places as well.

8) We noted that you cited Tables 1 and 2 as Supplementary Material; however, you labeled them as Tables 1 and 2. If the tables are supplementary ones , please label them as Table S1 and Table S2 . Please also remove them from the main file of the manuscript and upload them as separate files with the item type 'Supporting Information'. Please also ensure to include legends for the supplementary tables after the reference list.

**Reviewers' comments:**

Reviewer's Responses to Questions

**Comments to the Authors:**

**Please note that one of the reviews is uploaded as an attachment.**

Reviewer #1: The study “A novel DNA repair-independent role for Gen nuclease in promoting unscheduled polyploidy cell proliferation” by Budzyk et. al. explores the use of Gen (Gen1/Yen1) in cell proliferation and DNA replication in polyploid cells. They show that Gen is not needed in diploid cells for preventing DNA damage and genome instability, even upon aphidicolin induced replication stress. However, Gen is essential to maintain polyploid cells. Deletion of Gen leads to smaller and slower dividing polyploid cells. Interestingly, overexpression of Gen in diploids and in polyploids leads to ectopic DNA damage. Finally, overexpression of Gen is similar to overexpressed nuclease dead Gen in polyploid cells, indicating that Gen has a nuclease-indepdendent role in maintaining polyploidy.

Overall, it’s a well written manuscript. I have a few concerns regarding the manuscript that I outline below.

1. Fig 1G and 3 E have multiple statistical tests within the same figure. The authors claim they used Mann-Whitney test.

a. They should mention if it is two-tailed or one-tailed.

b. Multiple hypothesis testing is missing, q-values should be calculated and reported.

2. Fig 4B – while it looks obvious, a Chi square goodness of fit test or a similar statistical test should be done to show that the differences in Ctrl, GenOE and Genmut are statistically different.

3. In the introduction, it is not clear what the link is between whole genome duplication and HR. The authors should consider re-writing the section just to better explain why HR is needed in polyploid cells.

4. Results – “Using a NB specific GAL4 driver (Worniu-GAL4), we depleted by RNA

interference (RNAi) Rad51, BLM, Mus81, SLX1 and Gen in polyploid brains. Rad51

is a major player in HR [9], while the remaining proteins function in Holiday Junction

(HJ) processing [12,30].”

BLM also functions in resection and initiation of HR.

5. Ctrl WT – this is jargon – please define usage.

6. The authors show that GenND leads to increased hH2Av in polyploid cells. What about diploids? This is a major control, that should be done.

Reviewer #2: In this study the Authors identified a novel, DNA repair-independent role of Gen nuclease in promoting unscheduled polyploidy cell proliferation. Traditionally, Gen is known for its role in DNA damage repair, specifically in Holliday junction resolution during homologous recombination. However, this study demonstrates that Gen has an unexpected function in regulating DNA replication rates and cell cycle progression in polyploid cells. This is a strong Article that can be published.

Abstract: 1. Please, emphasize the unexpected, DNA repair-independent role of Gen nuclease earlier to immediately capture the reader’s attention. Please, strengthen the conclusion.

Introduction: 1.The introduction should smoothly transition from genome integrity and DNA damage response (DDR) to unscheduled whole genome duplication (WGD) and its implications. Currently, the link between WGD, replication stress, and the necessity for DNA damage repair is implied but not clearly stated. Make the connection more explicit. The final paragraph could more clearly emphasize the knowledge gap you are addressing and how your study contributes. 2. Please, explain benefits of your model using Drosophila neural stem cells- also known as neuroblasts (NBs) that are highly polyploid. It is also would be good to underline the important role of polyploidy in neurodegenerative disorders.

Discussion: 1. Please, clarify Gen’s Role in Diploid Cells; 2. Explain how loss-of-function delays mitosis, while overexpression accelerates it. 3. Highlight mitotic defects (e.g., chromosome bridges, lagging chromosomes). 4. Highlight that Gen is essential for polyploid cell proliferation, with loss-of-function reducing cell size and overexpression increasing proliferation despite DNA damage. 5. Discuss how cell cycle asynchrony and DNA replication rates are affected. 6. Please, discuss therapeutic and Biomedical Implications Emphasize that Gen inhibition selectively affects polyploid cells, making it a potential therapeutic target for polyploid tumors. 7. Propose future directions, including mammalian validation studies and drug development targeting Gen activity.

Reviewer #3: Review is uploaded as attachment.

**Have all data underlying the figures and results presented in the manuscript been provided?**

Reviewer #1: Yes

Reviewer #2: Yes

Reviewer #3: Yes

PLOS authors have the option to publish the peer review history of their article (what does this mean? ). If published, this will include your full peer review and any attached files.

**Do you want your identity to be public for this peer review?** For information about this choice, including consent withdrawal, please see our Privacy Policy .

Reviewer #1: No

Reviewer #2: **Yes: ** Olga Anatskaya

Reviewer #3: No

**Figure resubmission:**
---

## [Decision Letter · Decision Letter 1]

11 Aug 2025

Dear Dr Basto,

We are pleased to inform you that your manuscript entitled "A novel DNA repair-independent role for Gen nuclease in promoting unscheduled polyploidy cell proliferation" has been editorially accepted for publication in PLOS Genetics. Congratulations!

Yours sincerely,

Louise Cheng

Academic Editor

PLOS Genetics

Fengwei Yu

Section Editor

PLOS Genetics

Aimée Dudley

Editor-in-Chief

PLOS Genetics

Anne Goriely

Editor-in-Chief

PLOS Genetics

Comments from the reviewers (if applicable):

Reviewer's Responses to Questions

**Comments to the Authors:**

Reviewer #1: The authors have addressed all my concerns adequately.

Reviewer #2: Tha Authors carefully addressed all comments. Tha Article can be punlished.

Reviewer #3: Thank you for thoroughly addressing all of my comments.

**Have all data underlying the figures and results presented in the manuscript been provided?**

Reviewer #1: Yes

Reviewer #2: Yes

Reviewer #3: Yes

PLOS authors have the option to publish the peer review history of their article (what does this mean? ). If published, this will include your full peer review and any attached files.

**Do you want your identity to be public for this peer review?** For information about this choice, including consent withdrawal, please see our Privacy Policy .

Reviewer #1: No

Reviewer #2: No

Reviewer #3: No

**Data Deposition**

http://datadryad.org/submit?journalID=pgenetics&manu=PGENETICS-D-25-00127R1

**Press Queries**

---

## [Editor Report · Acceptance letter]

PGENETICS-D-25-00127R1

A novel DNA repair-independent role for Gen nuclease in promoting unscheduled polyploidy cell proliferation

Dear Dr Basto,

We are pleased to inform you that your manuscript entitled " 

A novel DNA repair-independent role for Gen nuclease in promoting unscheduled polyploidy cell proliferation" has been formally accepted for publication in PLOS Genetics! Your manuscript is now with our production department and you will be notified of the publication date in due course.

With kind regards,

Zsofia Freund

PLOS Genetics

On behalf of:
